# Disturbing the Redox Balance Using Buthionine Sulfoximine Radiosensitized Somatostatin Receptor-2 Expressing Pre-Clinical Models to Peptide Receptor Radionuclide Therapy with ^177^Lu-DOTATATE

**DOI:** 10.3390/cancers15082332

**Published:** 2023-04-17

**Authors:** Wendy Delbart, Gwennaëlle Marin, Basile Stamatopoulos, Roland de Wind, Nicolas Sirtaine, Pieter Demetter, Marie Vercruyssen, Erwin Woff, Ioannis Karfis, Ghanem E. Ghanem, Patrick Flamen, Zéna Wimana

**Affiliations:** 1Nuclear Medicine Department, Institut Jules Bordet, Hôpital Universitaire de Bruxelles (H.U.B), Université Libre de Bruxelles (ULB), 1000 Brussels, Belgium; 2Laboratory of Oncology and Experimental Surgery, Institut Jules Bordet, Hôpital Universitaire de Bruxelles (H.U.B), Université Libre de Bruxelles (ULB), 1000 Brussels, Belgium; 3Medical Physics Department, Institut Jules Bordet, Hôpital Universitaire de Bruxelles (H.U.B), Université Libre de Bruxelles (ULB), 1000 Brussels, Belgium; 4Laboratory of Clinical Cell Therapy, Institut Jules Bordet, Hôpital Universitaire de Bruxelles (H.U.B), Université Libre de Bruxelles (ULB), 1000 Brussels, Belgium; 5Pathology Department, Institut Jules Bordet, Hôpital Universitaire de Bruxelles (H.U.B), Université Libre de Bruxelles (ULB), 1000 Brussels, Belgium; 6Haematology Department, Institut Jules Bordet, Hôpital Universitaire de Bruxelles (H.U.B), Université Libre de Bruxelles (ULB), 1000 Brussels, Belgium

**Keywords:** peptide receptor radionuclide therapy, ^177^Lu-DOTATATE, radiosensitization, antioxidant defenses, BSO, glutathione, multiple myeloma

## Abstract

**Simple Summary:**

Peptide receptor radionuclide therapy with ^177^Lu-DOTATATE is an efficient treatment for patients suffering from metastasized neuroendocrine tumours. Nevertheless, suboptimal effects have been observed in the majority of patients. Hence, strategies to improve ^177^Lu-DOTATATE efficacy are desirable. Lu-177 induces oxidative stress, eventually leading to tumour cell death. Inhibition of the antioxidant defence mechanisms, using buthionine sulfoximine (BSO), represents an attractive strategy to increase ^177^Lu-DOTATATE efficacy. In cells and an animal model, the combination of ^177^Lu-DOTATATE and BSO was more effective than ^177^Lu-DOTATATE alone. In addition, it did not result in additional toxicity. Targeting the antioxidant defence system opens new safe treatment combination opportunities with ^177^Lu-DOTATATE.

**Abstract:**

Peptide receptor radionuclide therapy with ^177^Lu-DOTATATE improves the outcome of patients with somatostatin receptor (SSTR)-expressing neuroendocrine tumours. Nevertheless, stable disease has been the main response pattern observed, with some rare complete responses. Lu-177 exerts about two-thirds of its biological effects via the indirect effects of ionizing radiation that generate reactive oxygen species, eventually leading to oxidative damage and cell death. This provides a rationale for targeting the antioxidant defence system in combination with ^177^Lu-DOTATATE. In the present study, the radiosensitizing potential and the safety of depleting glutathione (GSH) levels using buthionine sulfoximine (BSO) during ^177^Lu-DOTATATE therapy were assessed in vitro and in vivo using a xenograft mouse model. In vitro, the combination resulted in a synergistic effect in cell lines exhibiting a BSO-mediated GSH decrease. In vivo, BSO neither influenced ^177^Lu-DOTATATE biodistribution nor induced liver, kidney or bone marrow toxicity. In terms of efficacy, the combination resulted in reduced tumour growth and metabolic activity. Our results showed that disturbing the cell redox balance using a GSH synthesis inhibitor increased ^177^Lu-DOTATATE efficacy without additional toxicity. Targeting the antioxidant defence system opens new safe treatment combination opportunities with ^177^Lu-DOTATATE.

## 1. Introduction

One of the major successes in the field of nuclear medicine has been the radiotheranostic approach in neuroendocrine tumours (NET) by selectively targeting somatostatin receptor subtype 2 (SSTR2) with the radiolabelled somatostatin analogue (SSA) ^177^Lu-DOTATATE. The landmark phase III randomized NETTER-1 clinical trial has validated the superiority of ^177^Lu-DOTATATE over long-acting SSAs in terms of progression-free survival and quality of life [1,2] and has led to its regulatory approval (i.e., Lutathera^®^), which consequently highly impacted NET patient’s management. Nevertheless, stable disease has been the main response pattern observed, with some rare complete responses (1% in the NETTER-1 trial) [1,3,4]. Diverse optimization strategies have therefore been investigated to improve patient outcomes after Lu-177-based radionuclide therapy. These strategies include dosimetry-based treatment individualization, new radiopharmaceuticals and combination therapies with radiosensitizing molecules [5,6,7,8,9], including the ones that target the DNA repair mechanisms [10,11,12]. Other potential targets for radiosensitization could be antioxidant defences. Cellular enzymatic and non-enzymatic detoxification systems exist to cope with excessive reactive species production from endogenous and exogenous sources, such as ionizing radiation (IR) [13]. As a β-emitter with a low linear energy transfer (LET), Lu-177 exerts about two-thirds of its biological effects via the indirect effects of IR that generate reactive oxygen species (ROS), eventually leading to oxidative damage and cell death. In our previous work, we have shown that antioxidant defences were significantly lower in cell lines with a higher sensitivity to ^177^Lu-DOTATATE compared to cell lines with a lower sensitivity to ^177^Lu-DOTATATE [14]. Consequently, targeting the antioxidant defences might represent a novel promising radiosensitizing strategy for peptide receptor radionuclide therapy (PRRT) with ^177^Lu-DOTATATE.

Among the variety of cellular antioxidants, glutathione (GSH) is a highly abundant thiol-containing tripeptide (L-glutamate, cysteine and glycine) that plays a prominent role in redox homeostasis. It acts directly as an oxidant scavenger and indirectly as a cofactor of several enzymes such as glutathione peroxidase, a first-line defence antioxidant [15]. Increased GSH levels have been implicated in tumour cell resistance to several chemotherapeutics (alkylating agents [16,17], platinum-containing compounds [18,19] and anthracyclines [20]) via the elimination of ROS, among other mechanisms [21,22]. Therefore, strategies to deplete intracellular GSH levels have been developed. L-Buthionine-(S,R)-Sulfoximine (BSO) is a potent and specific inhibitor of γ-glutamylcysteine synthetase (γ-GCS) [23], the rate-limiting factor in the biosynthesis of GSH. BSO causes intracellular GSH depletion by preventing its de novo synthesis [24,25], thereby enhancing sensitivity to various anti-cancer agents in vitro and in vivo [26,27]. In the clinical setting, BSO was mainly studied in combination with the alkylating agent melphalan [25].

In this study, we assessed the radiosensitizing potential of depleting GSH using BSO in combination with ^177^Lu-DOTATATE in vitro using a panel of human cancer cell lines expressing SSTR as well as in vivo using a multiple myeloma xenograft mouse model. Indeed, several evidence from the literature on SSTR expression in multiple myeloma cell lines [14,28,29] and patients [30,31,32,33] led us to consider multiple myeloma as a potential new indication for PRRT with ^177^Lu-DOTATATE. Our results showed that disturbing the cell redox balance using a GSH synthesis inhibitor increased ^177^Lu-DOTATATE efficacy without additional toxicity.

## 2. Materials and Methods

### 2.1. Cell Lines and Cell Culture

The melanoma cell lines (HBL [34,35] and MM162 [36]) were established in our laboratory. Multiple myeloma (COLO-677 and EJM), gastroenteropancreatic (GEP) (pancreatic adenocarcinoma, MIA-PACA-2 and colon adenocarcinoma, HT-29) cell lines were purchased from DSMZ (Braunschweig, Germany). HBL and MM162 were cultured in Ham’s F10 (Lonza, Basel, Switzerland); COLO-677 and HT-29 were cultured in RPMI-1640 (Sigma, St. Louis, MO, USA); EJM was cultured in Iscove’s MDM (Gibco, Invitrogen, Waltham, MA, USA) and MIA-PACA-2 was cultured in DMEM (Sigma). Media were supplemented with 10% or 20% (EJM) foetal bovine serum as well as L-glutamine (Sigma), penicillin (Sigma), streptomycin (Gibco, Invitrogen) and kanamycin (Bio Basic, Markham, ON, Canada) at standard concentrations. Cells were grown at 37 °C in a humidified 95% air and 5% CO_2_ atmosphere. All cell lines were regularly checked for mycoplasma contamination using a MycoAlert^®^ Mycoplasma Detection Kit (Lonza). Cell line authentication was performed with a short tandem repeat (STR) test (Eurofins Genomics, Germany). Cell lines were chosen for their SSTR expression [14] and their range of intrinsic radiosensitivities (to external beam radiation therapy [37]).

### 2.2. Production of ^177^Lu-DOTATATE 

The radiopharmacy facility in the nuclear medicine department at Institut Jules Bordet produced ^177^Lu-DOTATATE for clinical use, as previously described by [38]. Labelling was carried out with a fully automated process using a synthesis module with disposable cassettes (Modular Pharmlab, Eckert & Ziegler, Hopkinton, MA, USA). Nine GBq of non-carrier added ^177^LuCl3 (EndolucinBeta^®^, Ph. Eur, ITM, Garching/Munich, Germany) and 150 µg DOTATATE (Bachem AG, Bubendorf, Switzerland) in sodium ascorbate/acetate buffer were heated for 20 min at 80 °C. The obtained raw radioactive solution was purified using solid-phase extraction on a C18 cartridge. The radiolabeled peptide was then eluted with 1 mL of 50% ethanol, followed by 19 mL of saline and subsequent sterile filtration over a 0.22 µm filter (included in the disposable cassette). All quality controls were performed according to the European Pharmacopoeia, allowing the conditional release of the radiopharmaceutical after appearance, pH, radiochemical purity (specification: >95%) and pyrogenicity testing, and subsequent final release after sterility results.

### 2.3. In Vitro Treatments

Cells were seeded in 12-well plates (Corning^®^ CellBIND^®^ Multiple Well Plate, Merck, Rahway, NJ, USA) at different densities (HBL and COLO-677: 1000 cells/well; MM162: 2000 cells/well; EJM: 4000 cells/well; MIA-PACA-2 and HT-29: 200 cells/well) and, at least 4 h later, pre-treated or not with BSO (10^−7^ M, L-buthionine sulfoximine, B2515, Merck). A low to non-toxic concentration of BSO was chosen to have minimal effect on cell survival as a monotherapy. The next day, cells were exposed to ^177^Lu-DOTATATE, as previously described [14]. Five MBq of ^177^Lu-DOTATATE were added in each well (four replicates) for 4 h. Subsequently, as well as three days later, the radioactive medium was removed and replaced with fresh medium with or without BSO. A crystal violet assay was performed on day 10.

The coefficient of drug interaction (CDI) was used to analyse the interactions between ^177^Lu-DOTATATE and BSO.
$$CDI=\frac{survival\%(A+B)}{survival\%\left(A\right)\times survival\%(B)}$$
where *A* is ^177^Lu-DOTATATE and *B* is BSO. CDI < 1 indicates synergism, CDI = 1 indicates additivity and CD > 1 indicates antagonism between BSO and ^177^Lu-DOTATATE.

### 2.4. Cell Survival Assay

Cell survival was assessed with crystal violet. The culture medium was removed, and cells were gently washed with phosphate-buffered saline (PBS), fixed in 1% glutaraldehyde (Merck) in PBS for 15 min and stained with 0.1% crystal violet (Sigma) in water for 30 min. The plates were washed under running tap water and subsequently lysed with 0.2% Triton X-100 (Roche, Basel, Switzerland) in water for 90 min. The associated absorbance was measured at 590 nm using a BioTek^®^ 800™ TS Absorbance Reader (Hong Kong).

### 2.5. Animals and Treatments

Mice experiments were approved by the animal ethics committee of Université Libre de Bruxelles (Comité d’Ethique du Bien-Etre Animal (CEBEA)) and performed according to European guidelines. Female athymic nude Foxn1^nu^ mice (Charles River) (age = 5 to 6 weeks) were subcutaneously injected in the right flank with 1 × 10^6^ EJM cells suspended in 100 µL of PBS and matrigel (1:1; Cultrex™ Basement Membrane Extract, Type 3, R&D Systems). The tumours were allowed to grow for 4 weeks before initiating treatments. At a tumour size of approximately 0.5 cm^3^, mice were randomized into 4 treatment groups of 10 mice: control, ^177^Lu-DOTATATE, BSO and ^177^Lu-DOTATATE + BSO. BSO (10 mM, L-buthionine sulfoximine, B2515, Merck) was administered via drinking water [39] for 3 weeks, starting the day before a single intravenous (IV) injection of 30 MBq ^177^Lu-DOTATATE or saline. All mice were monitored three times a week for body weight and tumour growth using a calliper. Tumour volume (mm^3^) was calculated using the formula $\frac{\pi }{6}$ × length × width × height [40]. The relative tumour volume was represented and calculated as (tumour volume on measured day)/(tumour volume measured at baseline). Mice were sacrificed using cervical dislocation under anaesthesia once tumours exceeded the maximal ethical volume. Another group of mice receiving ^177^Lu-DOTATATE alone or in combination with BSO was sacrificed 24, 72 and 168 h post-injection (p.i.) for ex vivo biodistribution studies and early glutathione quantification (4 mice per group per time point) (Figure 1).

### 2.6. Ex Vivo Biodistribution Studies and Time-Activity Curves

The amount of radioactivity was measured in different organs at 24, 72 and 168 h p.i. Animals were sacrificed, and then the organs were collected and weighed, and radioactivity was measured using a gamma counter (2480 WIZARD2, Perkin Elmer, Waltham, MA, USA). The biodistribution data were used to determine the kinetics of the activity in organs. Tissue uptake was calculated as the percentage of injected activity per gram of tissue (%IA/g). The measured activity data as a function of time were fitted with a single exponential curve using the least-square regression method with GraphPad Prism version 7.01.

### 2.7. Glutathione Quantification

Total glutathione was assessed in cell lines and mice tissues using the Quantification kit for oxidized and reduced glutathione (Sigma), according to the manufacturer’s instructions. Cells in culture were harvested, washed and then lysed with freeze-thaw cycles and subsequently mixed with 5% 5-sulfosalicylic acid. After centrifugation at 8000× *g* for 10 min, the supernatant was collected, and the 5-sulfosalicylic acid concentration was reduced to 0.5% with water for glutathione quantification. Tumour, liver and kidneys were harvested, snap-frozen in liquid nitrogen and stored at −70 °C until assayed for glutathione. Tissues were homogenized in 5% 5-sulfosalicylic acid in lysing matrix D tubes (MP Biomedicals) containing ceramic beads using a FastPrep-24 homogenizer (MP Biomedicals, Irvine, CA, USA) and centrifuged at 8000× *g* for 10 min. The supernatant was collected, and the 5-sulfosalicylic acid concentration was reduced to 0.5% with water for glutathione quantification. The buffer solution was added to samples (either from cell lines or tissues) (in triplicates) in a 96-well plate. After 1 h incubation at 37 °C, the substrate (5,5′-dithiobis (2-nitrobenzoic acid)) (DTNB), coenzyme and enzyme (glutathione reductase) were added into the wells, followed by 10 min incubation at 37 °C. Absorbance was read at 405 nm using a microplate reader (Thermo Labsystems Multiskan EX, ThermoFisher, Waltham, MA, USA). Total glutathione concentration was determined using a calibration curve.

### 2.8. In Vivo Imaging and ^18^F-FDG PET/CT Images Analysis

Baseline ^18^F-FDG PET/CT imaging was performed 4 to 5 days before ^177^Lu-DOTATATE injection and repeated 3 weeks p.i. to assess metabolic tumour response. Studies were conducted using a μPET-CT (nanoScan PET-CT, Mediso, Budapest, Hungary) on all mice (40 mice, 10 per group) at baseline and on the remaining mice 17 days post-treatment (25 mice, 4–7 mice per group). First, ^18^F-FDG was injected in the tail vein, and then the mice were imaged 1 h p.i. Animals were scanned under isoflurane anaesthesia (induction: 3% isoflurane, 3L O_2_; maintenance: 1.5% isoflurane, 1.5 L O_2_) and kept at 37 °C using a thermoregulation unit (Minerve) during image acquisition. PET/CT image analysis and quantification were performed using MIM software. A three-dimensional anatomical contour of the tumour (volume of interest (VOI)) was manually drawn on the merged PET/CT images, and a threshold of 15% of the VOI maximum uptake value was applied to assess tracer uptake. To quantify radioactivity within the tumour, corresponding PET uptake values were normalized to the injected radioactivity per body weight. Changes in ^18^F-FDG uptake before and after treatment are represented as fractional increase (value-baseline/baseline) in total lesion glycolysis (TLG) (TLG = metabolic tumour volume × SUVmean) (ΔTLG). Changes in anatomical tumour volume before and after treatment are represented as fractional increase (value-baseline/baseline) in tumour volume (ΔVol_T_).

### 2.9. Ex Vivo Tissue Preparation and Hematoxylin and Eosin (H & E) Staining

Liver and kidneys were harvested, fixed in PAXgene Tissue FIX (Qiagen, Hilden, Germany) for 1 day at room temperature and kept at 4 °C in PAXgene Tissue STABILIZER (Qiagen) until they were embedded in paraffin for histological studies. Staining was carried out on 4 μm paraffin-embedded tissue sections subsequently deparaffinized in xylene and rehydrated in decreasing alcohol concentrations. Slides were incubated in Mayer’s hematoxylin for 6 min and subsequently in eosin for 3 min. Slides were mounted using Tissue-Tek Prisma (Sakura, Alphen aan den Rijn, The Netherlands). Digital scans of slides were acquired with Nanozoomer NDP (Hamamatsu Photonics, Shizuoka, Japan) and analysed by an experienced clinical pathologist.

### 2.10. Bone Marrow Assessment

At the time of sacrifice, bone marrow (BM) was harvested from both femurs (containing hematopoietically active marrow) of the animals. Femurs were excised from each mouse, and then the attached muscle was trimmed from the periosteal surface and the bone was opened at the distal end. BM was isolated using the cytocentrifuge method [41]. In brief, the femur was placed in a 0.5 mL Eppendorf tube pre-punctured with a needle at the base of the tube to form an opening. The 0.5 mL tube was nested within a 1.5 mL Eppendorf tube, and both were centrifuged at 10,000 rpm for 15 s to form a BM pellet at the bottom of the 1.5 mL Eppendorf tube yet to retain the femur within the 0.5 mL tube.
Bone marrow smears and cytological assessment.

BM smears were prepared using the push-slide technique [42]. In brief, one drop of BM was placed at one extremity of a glass slide and was then spread in an even film using a second glass slide (held at an angle) as a spreader. The spreader slide was slid forward the length of the slide to produce a smear. Slides were fixed in absolute methanol for 5 min and then air-dried. May-Grunwald–Giemsa staining was used for routine evaluation of BM cytomorphology. Smears were incubated for 10 min in the May-Grunwald stain (50%) diluted with pH 6.8 water, followed by a 25 min incubation step in the Giemsa stain (10%) diluted with pH 6.8 water. Slides were then rinsed 3 times with pH 6.8 water before being air-dried. A qualitative and quantitative cytological evaluation of all the major lineages was performed by an experienced clinical pathologist on 300 to 500 cells [42].
Bone marrow apoptosis assessment.

BM samples were assayed for apoptosis. A 10 min incubation step at 37 °C with NH4Cl was performed for red blood cell lysis. Samples were then centrifuged for 5 min, and the supernatant containing lysed red blood cells was discarded. Samples were then incubated with Annexin-V Binding Buffer (ThermoFisher, Waltham, MA, USA), Annexin V (BD Biosciences, Franklin Lakes, NJ, USA) and 7-AAD (BD Biosciences, Franklin Lakes, NJ, USA) for 15 min at room temperature. Annexin Buffer was added to samples that were subsequently analysed using a MACSQuant Analyzer 10 Flow Cytometer (Miltenyi Biotec, Bergisch Gladbach, Germany).

### 2.11. Statistical Analyses

Statistical analyses were performed with GraphPad Prism 7.01 (GraphPad Software). After normality verification using the Shapiro–Wilk test, a parametric *t*-test was performed when comparing 2 means. A two-way ANOVA with a Turkey’s multiple comparison test was used to assess differences in apoptotic stages between treatment groups. A one-way ANOVA or Kruskal–Wallis test was used to assess differences in animal groups. Outliers were identified using the Grubb’s method and excluded from statistical analysis.

## 3. Results

### 3.1. BSO-Radiosensitized SSTR-Positive Human Cancer Cell Lines to ^177^Lu-DOTATATE

SSTR-expressing cell lines [14] were used to assess the radiosensitizing potential of BSO in combination with ^177^Lu-DOTATATE. BSO alone did not affect cell lines survival except in HBL (81.2 ± 3.8%, *p* = 0.002) and MIA-PACA-2 (87.6 ± 3.8%, *p* = 0.004). In contrast, ^177^Lu-DOTATATE alone significantly decreased cell survival in all cell lines compared to the control: HBL (67.3 ± 2.0%, *p* < 0.001), MM162 (86.2 ± 1.9%, *p* < 0.001), COLO-677 (67.1 ± 2.0%, *p* < 0.001), EJM (77.8 ± 1.2%, *p* < 0.001), MIA-PACA-2 (80.8 ± 2.4%, *p* < 0.001) and HT-29 (93.1 ± 1.8%, *p* = 0.004). When combined with BSO, cell survival was further reduced in HBL (48.9 ± 5.3%, *p* = 0.002), MM162 (75.2 ± 2.2%, *p* = 0.001), EJM (63.5 ± 1.5%, *p* < 0.001) and MIA-PACA-2 (62.2 ± 3.6%, *p* < 0.001) but not in COLO-677 (63.2 ± 1.3%, *p* = 0.16) and HT-29 (91.4 ± 1.6%, *p* = 0.49) (Figure 2). The combination resulted in a synergistic effect (CDI < 1) in all cell lines affected by the treatment combination (Table 1).

The BSO exposure one day before ^177^Lu-DOTATATE treatment was able to decrease glutathione levels in all cell lines except COLO-677 and HT-29, which were not affected (HBL: −21.5 ± 3.0%, *p* < 0.001; MM162: −11.1 ± 3.7%, *p* = 0.02; EJM: −9.4 ± 2.2%, *p* = 0.001 and MIA-PACA-2: −20.7 ± 3.5%, *p* < 0.001) (Figure 3).

### 3.2. BSO Did Not Influence ^177^Lu-DOTATATE Biodistribution nor Pharmacokinetics In Vivo

An ex vivo biodistribution study was performed to evaluate the impact of BSO on ^177^Lu-DOTATATE body distribution. The activity concentrations in normal tissues are shown in Appendix A. Twenty-four hours p.i., the highest uptake of ^177^Lu-DOTATE alone or in combination was found in kidneys and urine, reflecting the known urinary excretion of ^177^Lu-DOTATATE as well as partial reabsorption of the radiotracer. In addition, ^177^Lu-DOTATATE accumulated in SSTR2-expressing organs such as the pancreas and stomach. No statistically significant differences in the ^177^Lu-DOTATATE uptake between the two treatment groups (^177^Lu-DOTATATE alone and its combination with BSO) were observed in any of the investigated organs/tissues and time points (Figure 4).

Time–activity curves were equivalent between both groups for the liver and kidneys, with overlapping 95% confidence bands, demonstrating similar pharmacokinetics (Figure 5). No differences in effective half-life between the groups were observed in the kidneys (^177^Lu-DOTATATE: 0.57 day; ^177^Lu-DOTATATE + BSO: 0.55 day) or liver (^177^Lu-DOTATATE: 1.00 day; ^177^Lu-DOTATATE + BSO: 0.94 day).

### 3.3. BSO Reduced Glutathione Levels in the Liver, Kidneys and Tumour

Total glutathione levels were checked in the liver, kidneys and tumour. Liver glutathione levels in the combination group were 30% lower than in the ^177^Lu-DOTATATE group, and this reduction was maintained over time (although, at the statistical limit on day 4): −26 ± 4%, *p* = 0.02 (day 2), −32 ± 11%, *p* = 0.06 (day 4), −31 ± 7%, *p* = 0.02 (day 8). On a longer-term basis (days 14–18), the difference did not reach statistical significance (−29 ± 13%, *p* = 0.12). In the kidneys, the glutathione levels were 49 ± 16% (*p* = 0.11) and 35 ± 8% (*p* = 0.02) less after 2 and 4 days of continuous BSO exposure, respectively, compared to ^177^Lu-DOTATATE alone. Although glutathione levels normalized compared to ^177^Lu-DOTATATE alone after 8 days (−1 ± 4%, *p* = 0.85), a further decrease occurred on days 14 to 18 (−74 ± 6%, *p* = 0.007) (Figure 6).

Tumour glutathione levels were efficiently depleted with a 2-day BSO exposure in the combination group compared to ^177^Lu-DOTATATE alone (−79 ± 8%, *p* < 0.001) (Figure 7A). After 14 to 18 days of continuous BSO exposure, total glutathione levels were maintained reduced in the combination group compared to ^177^Lu-DOTATATE alone (−85 ± 1%, *p* = 0.006). Although ^177^Lu-DOTATATE alone decreased glutathione levels compared to the control group (−36 ± 10%, *p* = 0.01), it did not further reduce glutathione levels in combination with BSO compared to BSO alone (−6 ± 10%, *p* < 0.99) (Figure 7B).

BSO was administered via drinking water; therefore, mice water consumption was monitored to verify it was not influenced by the presence of BSO. There were no significant differences in water intake between the different treatment groups: control: 5.0 ± 0.6 mL/day/mouse; BSO: 5.6 ± 0.6 mL/day/mouse; ^177^Lu-DOTATATE: 5.2 ± 0.5 mL/day/mouse; and ^177^Lu-DOTATATE + BSO: 5.1 ± 0.7 mL/day/mouse) (Appendix A). This resulted in a BSO dose of 2 mmol/kg/day.

### 3.4. The Combination of ^177^Lu-DOTATATE with BSO Did Not Result in Signs of Additional Hemato-, Nephro- or Hepatotoxicity

The evaluated hematotoxicity in mice sacrificed post-^177^Lu-DOTATATE injection (29 to 38 days), revealed no significant difference in the number of BM apoptotic cells among the different treatment groups (control: 28.6 ± 3.2%; ^177^Lu-DOTATATE: 33.0 ± 2.8%; ^177^Lu-DOTATATE + BSO: 30.6 ± 2.6%) (Figure 8A). Furthermore, the assessment of the relative proportion of the different BM lineages revealed no significant differences between treatment groups for any of the erythroid, granulocytic or lymphoid lineages (Figure 8B–D). In addition, the qualitative assessment of BM smears did not show any morphological abnormalities in the major BM lineages, including the megakaryocytes population.

Analysis of the H&E-stained slides of the kidneys revealed no morphological alterations and no inflammatory infiltrates in the renal cortex of any of the mice of the four treatment groups (Figure 9). Similarly, the liver tissues showed no signs of toxicity (Figure 10).

### 3.5. The Combination of ^177^Lu-DOTATATE with BSO Reduced Tumour Growth

The efficacy of ^177^Lu-DOTATATE (mean injected activity: 29.5 ± 0.4 MBq) and its combination with BSO was assessed using the EJM multiple myeloma xenograft model. The mean tumour size at the time of treatment administration was 487 ± 295 mm^3^, with no significant differences in the tumour volume between treatment groups. Longitudinal assessment of tumour growth revealed an increase in subcutaneous tumour volume in all treatment groups with different tumour growth rates (Figure 11A,C). Tumour growth slowed down with an increased doubling time induced with ^177^Lu-DOTATATE alone (9.8 days) and further enhanced in combination with BSO (12.4 days) compared to the control (7.4 days) or BSO alone (7.5 days) (Figure 11C). Three weeks after the start of the treatment (day 45 post-inoculation), the relative tumour volume was significantly lower in the combination group (4.4, IQR: 2.7) compared to the control (8.4, IQR: 6.2, *p* = 0.03) and BSO (7.6, IQR: 7.3, *p* = 0.02) groups but not to ^177^Lu-DOTATATE alone (6.1, IQR: 5.3, *p* = 0.32) (Figure 11B). This was corroborated by a significantly smaller ΔVol_T_, measured with CT, in the combination group compared to the control group (4.5, IQR: 3.8 versus 11.1, IQR: 8.3, *p* = 0.04), but not compared to the other groups (BSO: 9.1, IQR: 9.3, *p* = 0.15; ^177^Lu-DOTATATE: 7.5, IQR: 11.9, *p* = 0.27) (Figure 11D).

### 3.6. The combination of ^177^Lu-DOTATATE with BSO Reduced ^18^F-FDG Metabolic Activity

The effects of ^177^Lu-DOTATATE and BSO on tumour metabolism were assessed using the changes in ^18^F-FDG uptake (mean injected activity: 4.44 ± 0.30 MBq). At baseline, there was no significant difference in TLG among the four treatment groups. A statistically significant lower ΔTLG was observed in the combination group (3.2, IQR: 1.9) compared to the control (8.5, IQR: 4.4, *p* = 0.02) and BSO (8.8, IQR: 9.3, *p* = 0.04) groups but not to ^177^Lu-DOTATATE alone (4.8, IQR: 7.9, *p* = 0.20) (Figure 12).

## 4. Discussion

^177^Lu-DOTATATE has significantly changed the therapeutic landscape of SSTR-expressing NETs, resulting in increased survival and quality of life for NET patients [1]. Nevertheless, complete responses are rare, and optimization strategies are being investigated to improve the outcome for ^177^Lu-DOTATATE-treated patients. Potentiating the effects of ^177^Lu-DOTATATE, through its combination with radiosensitizing molecules, appears to represent a promising approach [5,6,7,8]. By the same token, targeting specific molecular pathways, based on a robust radiobiology rationale, could result in desired synergistic effects. In this study, we assessed the radiosensitizing potential of GSH depletion using BSO both in vitro, using different cancer cell lines, and in vivo, using an SSTR-expressing multiple myeloma xenograft model. The multiple myeloma model was chosen for the reason that it is a radiosensitive malignancy expressing SSTR that could allow us to consider it as a novel indication for PRRT.

In our SSTR-expressing human cancer cell line panel, BSO had a synergistic effect when combined with ^177^Lu-DOTATATE (Table 1), which is conceivably explained by the decreased glutathione levels (up to −20%) (Figure 3). Conversely, in cells without a significant decrease in glutathione levels, the radiosensitization effect was absent (Figure 2). Similarly, in vivo, in our multiple myeloma model, BSO-mediated radiosensitization to ^177^Lu-DOTATATE resulted in a significant decrease in tumour growth rate, tumour volume and tumour 18F-FDG uptake compared to the control and BSO groups (Figure 11 and Figure 12). Once more, the observed BSO radiosensitizing effect was associated with glutathione depletion under BSO administration (Figure 7).

The rationale to target GSH was based on our previous work suggesting a key role for cellular antioxidant defences in ^177^Lu-DOTATATE radioresistance [14], and BSO was chosen for its potency in the inhibition of GSH as well as its favourable toxicity profile. The current study confirms the potency of BSO with tumour GSH depletion of ~80%, which is consistent with studies using comparable BSO dosing schedules in mice resulting in an 80 to 95% reduction in various tumour types [22,43]. In terms of toxicity, in our in vivo model, BSO (2 mmol/kg/day) was well tolerated and did not lead to water consumption differences between the treatment groups (Appendix A), which are consistent with similar observations from other groups [39,44]. Of note, in the clinical setting, early-phase clinical studies on solid tumours (melanoma, ovarian, pancreas, breast, colon, renal and lung cancer) appeared safe, with only grade 1–2 nausea and vomiting caused by BSO alone [45,46,47,48]. Nevertheless, combined with melphalan, grade 3–4 myelosuppression was observed in heavily pretreated patients [48]. Whether this degree of myelosuppression was related to the addition of BSO was uncertain [25]. In a phase I trial, BSO was shown to significantly impact melphalan pharmacokinetics, with decreased melphalan clearance and volume of distribution compared to melphalan given alone [46]. In our study, BSO administration did not result in a change in ^177^Lu-DOTATATE biodistribution and pharmacokinetics nor increased toxicity of the combination compared to ^177^Lu-DOTATATE alone. Indeed, the results of the biodistribution of ^177^Lu-DOTATATE were in line with previous studies [49,50] and were not affected by BSO (Figure 4 and Figure 5, Appendix A).

Although ^177^Lu-DOTATATE is a well-tolerated treatment with moderate toxicity, side effects may occur mainly in the BM and to a much lesser degree in the kidneys [51,52], making them the organs at risk (OAR). Therefore, BSO may potentially radiosensitize these OAR and consequently exacerbate ^177^Lu-DOTATATE-associated adverse events. Furthermore, while the liver is not fundamentally considered as an OAR for ^177^Lu-DOTATATE treatment alone, when combined with BSO, due to the critical role of GSH in detoxification functions within the hepatic cells, its depletion may induce ^177^Lu-DOTATATE radiosensitization as well as impact normal hepatic function, potentially turning the liver into an AOR. As also reported by other groups [24,39,53], we demonstrated that BSO decreased kidney and liver glutathione levels (Figure 6), with a greater decrease for the former reflecting the faster turnover of GSH in the kidneys. Of note, the GSH level fluctuation over time observed in our results may be related to GSH circadian rhythm [39,54]. Nevertheless, at the time of sacrifice, no signs of kidney (Figure 9) or liver (Figure 10) toxicity were observed in any of the treatment groups, even in the absence of a nephroprotective solution. Our results are consistent with the preclinical findings of Sun et al., who investigated a series of biochemical indicators of liver toxicity and found no differences in mice that received 30 mM BSO via drinking water compared to the control, suggesting that BSO was an effective and nontoxic method for lowering GSH levels [44]. Additionally, Skapek et al. showed that BSO-mediated GSH depletion prior to treatment with melphalan did not result in enhanced organ toxicity compared to melphalan alone in nude mice [55]. Their BSO dosing schedule consisted of seven doses of intraperitoneal BSO injections (2.5 mmol/kg) at 12 h intervals and concomitant oral administration via drinking water (20 mM), an even higher dose of BSO compared to our study. Furthermore, the Bailey et al. study on 28 patients receiving BSO alone or in combination with melphalan did not report any renal or hepatic toxicity [46].

The most common toxicity with ^177^Lu-DOTATATE is haematological, with about 10% of patients developing grade 3/4 subacute haematological toxicities [1,3,52]. In our study, no BM toxicity was observed in the ^177^Lu-DOTATATE group, and no negative impact of the addition of BSO on the hematopoietic system was observed, as shown by the preserved apoptosis levels, cell morphology and cell count. Except for some BM samples, the percentages of the major BM lineages were indeed within reported ranges for C57BL/6J mice [56] (Figure 8). Albeit the performed evaluation of BM was partial and not fully differential, by grouping several developmental stages, it is considered adequate to characterize the major cytological findings in the BM [42]. Furthermore, Svensson et al. showed no acute or long term (sacrifice at 6 months p.i.) BM toxicity in nude mice treated with higher activities of ^177^Lu-DOTATATE (90 MBq), as measured by leucocyte count [57]. In our study, the follow-up time for toxicity assessment was limited, and possible long-term adverse effects may be encountered. Additionally, one should be cautious interpreting BM toxicity results in the preclinical setting as commonly used models (including our model) do not receive prior lines of treatments as those mainly found in the clinical setting. Furthermore, for multiple myeloma patients, an increased BM toxicity might be expected as a result of limited BM reserves (from previous sequential myelotoxic treatments) but also of higher BM absorbed doses of ^177^Lu-DOTATATE (due to the intramedullary localisation of lesions).

Apart from being safe, the combination of PRRT and BSO was shown to be effective, achieving a significant decrease in tumour growth rate, tumour volume and tumour 18F-FDG uptake compared to the control and BSO but not to ^177^Lu-DOTATATE alone (Figure 11 and Figure 12). The low therapeutic effect of ^177^Lu-DOTATATE in our in vivo mouse model could be justified by the fact that animals received a single administration of the radiotracer (as opposed to up to four injections in clinical practice) and that the mean tumour volume at baseline was quite high, with a great variation within groups (487 ± 295 mm^3^). For the latter, different groups have shown the influence of tumour size on PRRT response, with the effects of 111In-DTPA-octreotide [58] or ^177^Lu-DOTATATE [59] being less pronounced in rats bearing large CA20948 pancreatic tumours (>8 cm^2^ for 111In-DTPA-octreotide and >1 cm^2^ for ^177^Lu-DOTATATE) compared to small ones (≤1 cm^2^). Several reasons were considered to explain this observation and could be relevant to our model, such as less accessible area (due to poor vascularization and enhanced interstitial pressure) or less effective treatment (due to hypoxic regions or loss of SSTR expression following tumour dedifferentiation). In addition, Schmitt et al. found an inverse linear relationship between tumour size and ^177^Lu-DOTATATE uptake in a small cell lung cancer mouse model [49]. Nevertheless, despite these drawbacks, a sub-lethal dose of BSO combined with a sub-lethal activity of ^177^Lu-DOTATATE led to a significant efficacy of the treatment combination, highlighting the radiosensitizing potential of BSO.

Although we investigated GSH depletion-mediated radiosensitization of ^177^Lu-DOTATATE using a multiple myeloma xenograft mouse model, the first application of the treatment combination in the clinic would likely be with the approved indication of ^177^Lu-DOTATATE, namely NETs. A superior radiosensitization effect may even be expected in NETs given their higher GSH levels and their relatively radioresistant nature compared to multiple myeloma [14]. Nevertheless, multiple myeloma could represent a novel indication for PRRT with ^177^Lu-DOTATATE. Indeed, several in vitro and in vivo studies, including our own, have reported the expression of functional SSTR in multiple myeloma [14,28,29,30,31,32]. We have launched a phase-II trial to assess SSTR2 expression by 68Ga-DOTATATE PET/CT in symptomatic relapsing and refractory multiple myeloma patients (SCARLET trial—NCT04379817). Potential tumour responses may be anticipated in this radiosensitive malignancy and should be assessed with a representative BM environment, which was lacking in our mouse model.

The optimal combination treatment schedule would still have to be determined, which would be the one allowing the use of the minimal effective dose of BSO, i.e., the dose necessary to achieve sufficient tumour GSH depletion to have an efficient tumour radiosensitization, without excessive healthy tissue toxicity. The rapid clearance of BSO (estimated half-life in plasma of less than 2 h [46]) and the rapid rate of GSH turnover [60] would allow one to closely control GSH levels with BSO and allow for a GSH depletion only during tumour irradiation with ^177^Lu-DOTATATE, limiting the potential toxicities.

## 5. Conclusions

In conclusion, we investigated for the first time the potential for disturbing the redox balance by decreasing GSH levels with BSO as a new radiosensitizing strategy for ^177^Lu-DOTATATE in vitro and in vivo. We provided promising efficacy and safety data to support clinical trials using BSO in combination with ^177^Lu-DOTATATE. Efficient tumour GSH depletion was achieved, without significant BSO-related toxicity, resulting in an increased therapeutic effect of ^177^Lu-DOTATATE. Furthermore, we provided the first therapeutic data about ^177^Lu-DOTATATE in a potential novel indication, multiple myeloma.

## Figures and Tables

**Figure 1 cancers-15-02332-f001:**
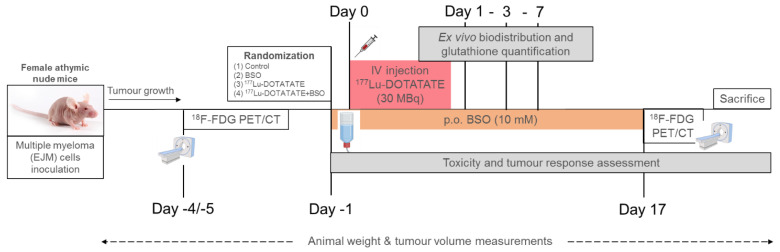
Timeline of the in vivo experiments. Female athymic nude mice were subcutaneously injected in the right flank with EJM multiple myeloma cells. Mice were randomized into four treatment groups: (1) control (IV saline injection), (2) BSO (IV saline injection and BSO via drinking water), (3) ^177^Lu-DOTATATE (a single IV injection of 30 MBq) and (4) ^177^Lu-DOTATATE + BSO (a single IV injection of 30 MBq and BSO via drinking water). Mice had a baseline and a post-treatment ^18^F-FDG PET/CT to assess metabolic tumour response. Tumour growth was also followed using a calliper. Another group of mice receiving ^177^Lu-DOTATATE alone or in combination with BSO was sacrificed on days 1, 3 and 7 p.i. for ex vivo biodistribution studies and early glutathione quantification. All animals were continuously monitored for body weight. IV = intravenous, p.o. = per os.

**Figure 2 cancers-15-02332-f002:**
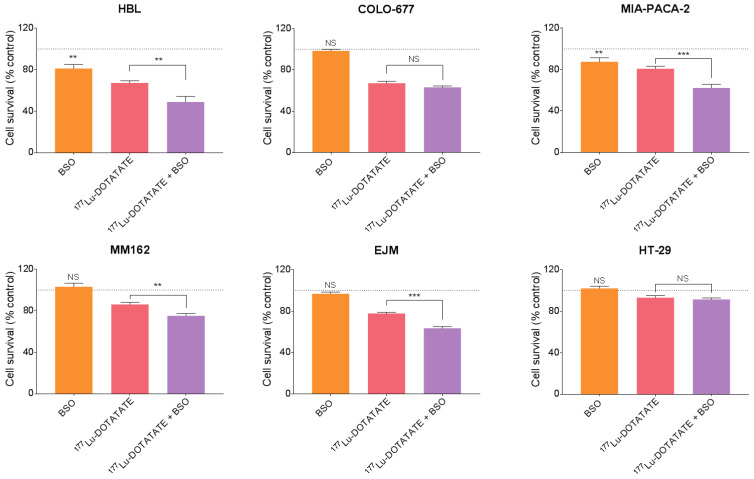
Effect of BSO and its combination with ^177^Lu-DOTATATE on the survival of melanoma (HBL and MM162), multiple myeloma (COLO-677 and EJM) and GEP (MIA-PACA-2 and HT-29) cell lines. Cells were exposed to 5 MBq of ^177^Lu-DOTATATE for 4 h. BSO (10^−7^M) was present in the medium from the day before irradiation until cell survival assessment on day 10. Results are expressed as a percentage of the non-treated counterpart and are represented as mean ± SEM (n = 12 from 3 independent experiments). The black dotted line represents 100%. *** *p* ≤ 0.001; ** *p* ≤ 0.01; NS = non-statistically significant.

**Figure 3 cancers-15-02332-f003:**
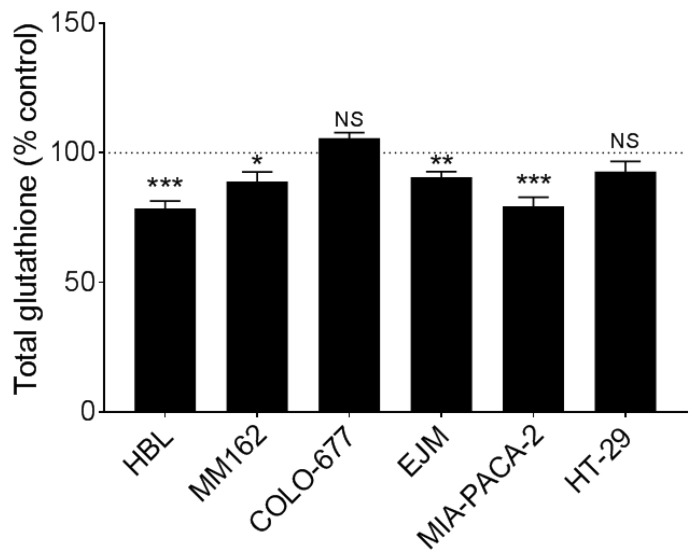
Total glutathione in melanoma (HBL and MM162), multiple myeloma (COLO-677 and EJM) and GEP (MIA-PACA-2 and HT-29) cell lines exposed to BSO. Total glutathione quantification was performed using cell lines exposed to 10^−7^ M BSO for 18 h. Results are expressed as a percentage of the non-treated counterpart (black dotted line) and are represented as mean ± SEM (n = 6 from 2 independent experiments). *** *p* ≤ 0.001; ** *p* ≤ 0.01; * *p* ≤ 0.05; NS = non-statistically significant.

**Figure 4 cancers-15-02332-f004:**
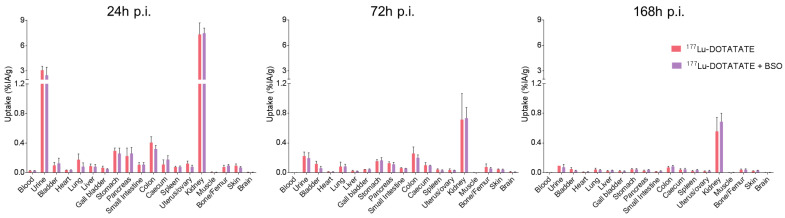
Biodistribution of ^177^Lu-DOTATATE at 24, 72 and 168 h p.i. in EJM xenograft mice. Mice in both groups were intravenously injected with 30 MBq of ^177^Lu-DOTATATE on day 0. Mice in the combination group (^177^Lu-DOTATATE + BSO) additionally received 10 mM BSO via drinking water, starting on day-1. Results are expressed as the mean of the % injected activity per gram of tissue (%IA/g) ± SD of 4 mice per group.

**Figure 5 cancers-15-02332-f005:**
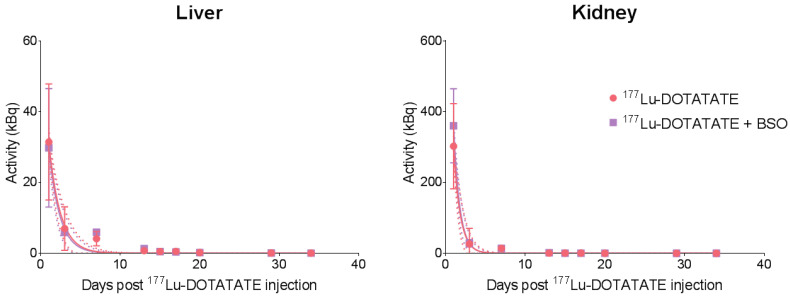
^177^Lu-DOTATATE time–activity curves of the liver and kidneys of EJM xenograft mice. Mice in both groups were intravenously injected with 30 MBq of ^177^Lu-DOTATATE on day 0. Mice in the combination group (^177^Lu-DOTATATE + BSO) additionally received 10 mM BSO via drinking water, starting on day-1. Solid lines represent the single exponential curve fitting. Dashed lines represent 95% confidence bands. n = 1 to 6 mice per group.

**Figure 6 cancers-15-02332-f006:**
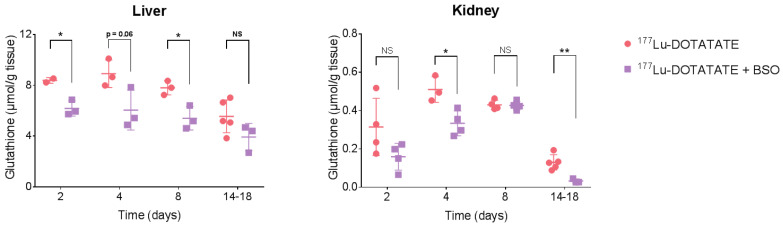
Effect of BSO on total glutathione levels in the liver and kidneys. Mice were intravenously injected with 30 MBq of ^177^Lu-DOTATATE on day 0. BSO (10 mM) was administered via drinking water starting on day-1. Total glutathione levels in the liver and kidneys after 2, 4, 8 and 14–18 days of continuous BSO exposure. Results are expressed as the mean ± SD of 2 to 5 mice per group (pooled data from mice sacrificed between days 14 and 18). ** *p* ≤ 0.01; * *p* ≤ 0.05; NS = non-statistically significant.

**Figure 7 cancers-15-02332-f007:**
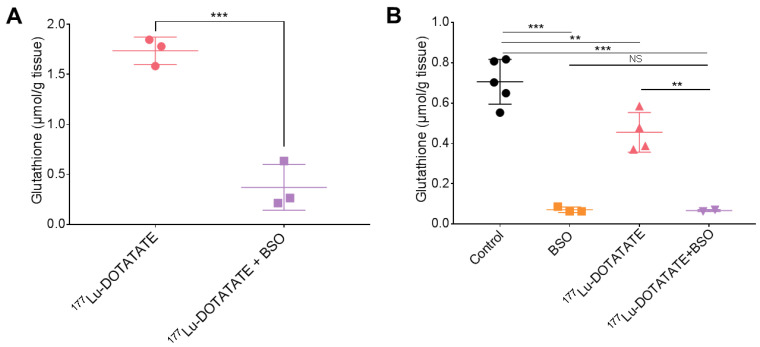
Effect of BSO on total glutathione levels in EJM tumours. Mice were intravenously injected with 30 MBq of ^177^Lu-DOTATATE on day 0. BSO (10 mM) was administered via drinking water starting on day-1. Total glutathione levels in tumours after 2 days (**A**) and 14–18 days (**B**) of continuous BSO exposure. Results are expressed as the mean ± SD of 2 to 5 mice per group (pooled data from mice sacrificed between days 14 and 18). *** *p* ≤ 0.001; ** *p* ≤ 0.01; NS = non-statistically significant.

**Figure 8 cancers-15-02332-f008:**
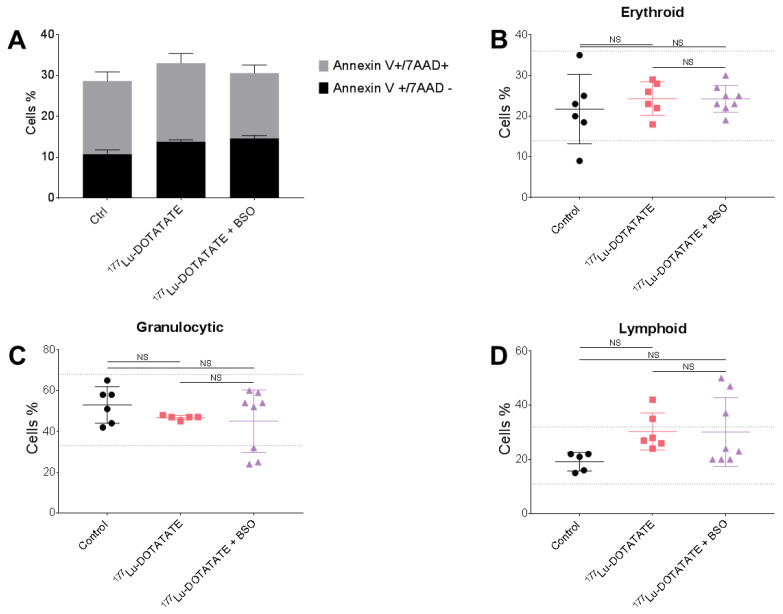
Effect of ^177^Lu-DOTATATE and its combination with BSO on bone marrow. Mice were intravenously injected with 30 MBq of ^177^Lu-DOTATATE on day 0. Mice in the combination group (^177^Lu-DOTATATE + BSO) additionally received 10 mM BSO via drinking water starting on day-1. BM (from both femurs) of 3 to 4 mice per group, sacrificed 29 to 38 days post-^177^Lu-DOTATATE injection, were analysed for apoptosis (**A**) or cytological assessment (**B**–**D**). (**A**) Apoptotic cells were classified into early-apoptotic (Annexin V+/7AAD−) and late-apoptotic/necrotic cells (Annexin V+/7AAD+). Results are expressed as mean ± SEM. The relative proportions of major BM lineages were evaluated: erythroid (**B**), granulocytic (**C**) and lymphoid lineages (**D**). Results are represented as mean ± SD. The dotted lines represent lower and upper normal ranges. NS = non-statistically significant.

**Figure 9 cancers-15-02332-f009:**
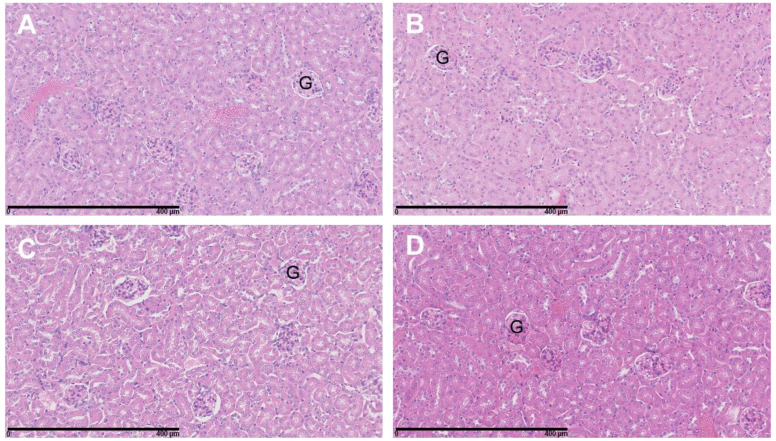
Representative H&E-stained slides of mice kidneys. (**A**) Control, (**B**) BSO (**C**), ^177^Lu-DOTATATE and (**D**) ^177^Lu-DOTATATE + BSO. Magnification 20×. No morphological changes were observed. G = glomeruli.

**Figure 10 cancers-15-02332-f010:**
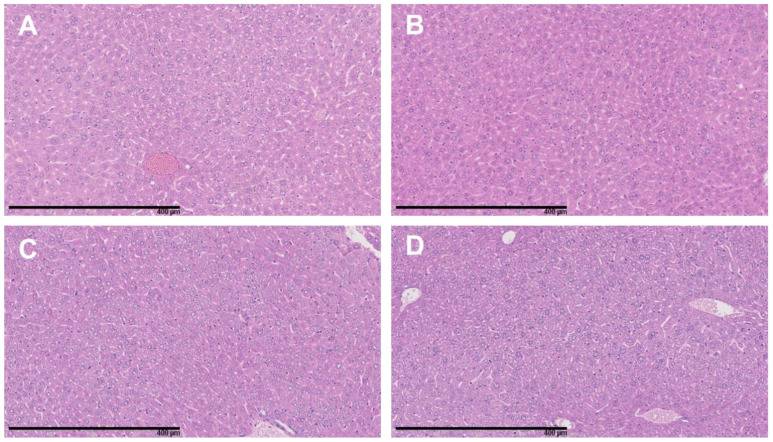
Representative H&E-stained slides of mice liver. (**A**) Control, (**B**) BSO, (**C**) ^177^Lu-DOTATATE and (**D**) ^177^Lu-DOTATATE + BSO. Magnification 20×. No morphological changes were observed.

**Figure 11 cancers-15-02332-f011:**
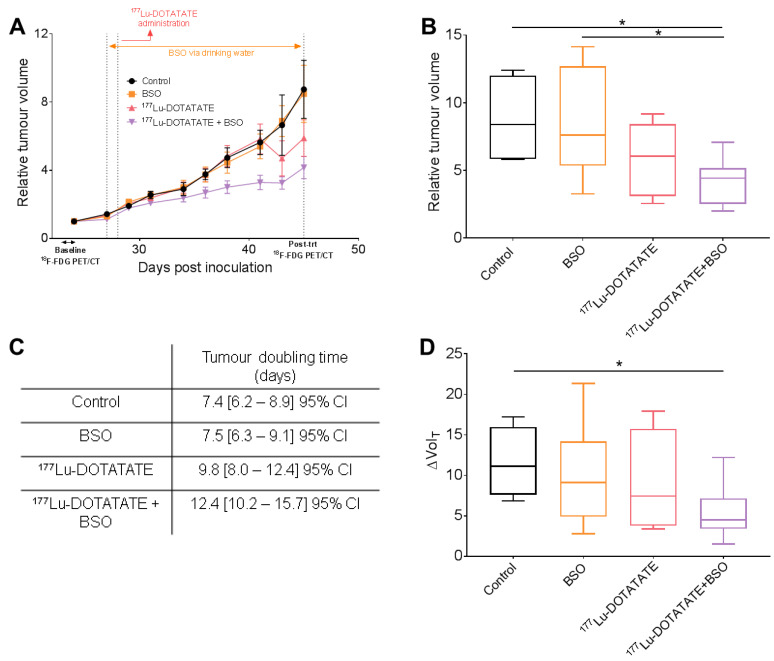
Effect of ^177^Lu-DOTATATE alone and in combination with BSO on EJM tumours. Mice bearing EJM multiple myeloma tumours were treated on day 28 post-tumour cell inoculation with 30 MBq of ^177^Lu-DOTATATE alone or in combination with 10 mM BSO via drinking water, starting on day 27, for 3 weeks. Tumours were measured 3 times per week in 8 to 10 mice per group. (**A**) Tumour growth over time, represented as mean relative tumour volume ± SEM. (**B**) Box plot distribution of relative tumour volume (calliper measurement) on day 45. Only significant differences are indicated, where * *p* ≤ 0.05. (**C**) Tumour doubling time was obtained by fitting an exponential growth equation from tumour growth measures. (**D**) Box plot distribution of changes in tumour volume before and after treatment, measured on the CT part of the PET/CT, represented as a fractional increase in tumour volume (ΔVol_T_). Only significant differences are indicated, where * *p* ≤ 0.05.

**Figure 12 cancers-15-02332-f012:**
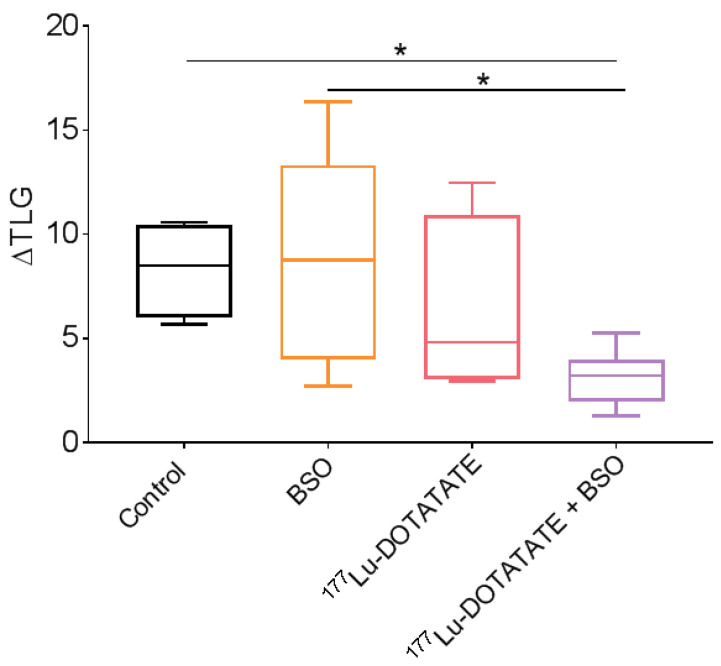
Effect of ^177^Lu-DOTATATE alone and in combination with BSO on EJM tumours ^18^F-FDG metabolic activity. Mice bearing EJM multiple myeloma tumours were treated on day 28 post-tumour cells inoculation with 30 MBq of ^177^Lu-DOTATATE alone or in combination with 10 mM BSO via drinking water, starting on day 27, for 3 weeks. Box plot distribution showing the changes in ^18^F-FDG uptake before and after treatment (day 45 post-inoculation) represented as fractional increase in total lesion glycolysis (ΔTLG). Only significant differences are indicated, where * *p* ≤ 0.05.

**Table 1 cancers-15-02332-t001:** The coefficient of drug interaction (CDI) for ^177^Lu-DOTATATE combined with BSO. CDI = survival%(A + B)/(survival%(A) × survival%(B)) where A is ^177^Lu-DOTATATE and B is BSO. CDI < 1 indicates synergism between BSO and ^177^Lu-DOTATATE. NA = not applicable.

HBL	MM162	COLO-677	EJM	MIA-PACA-2	HT-29
0.89	0.83	NA	0.85	0.88	NA

## Data Availability

The data presented in this study are available in this article.

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
