# Peer review of "Disturbing the Redox Balance Using Buthionine Sulfoximine Radiosensitized Somatostatin Receptor-2 Expressing Pre-Clinical Models to Peptide Receptor Radionuclide Therapy with 177Lu-DOTATATE"

_cancers, 2023, doi:10.3390/cancers15082332_

Round 1

Reviewer 1 Report

The authors must be commended about this research work. About the study design, I would have liked to include neuroendocrine tumor cell lines and assessment of response using somatostatin receptor PET/CT, although SSTR are known to be expressed by Multiple myeloma. Having said that, the authors have made a scientifically sound case for translational clinical trials using BSO in molecular radiotherapy.

Author Response

I would like to thank reviewer 1 for his comments about our research article.

Although MIA-PACA-2 and HT-29 are not NETs, they are cell lines with neuroendocrine differentiation and can be used as models for research on PRRT (Gradiz R, Silva HC, Carvalho L, Botelho MF, Mota-Pinto A. MIA PaCa-2 and PANC-1 - pancreas ductal adenocarcinoma cell lines with neuroendocrine differentiation and somatostatin receptors. Sci Rep. 2016 Feb 17;6:21648. doi: 10.1038/srep21648. PMID: 26884312; PMCID: PMC4756684.)

The treatment combination has been tested on these cell lines in vitro, but not in vivo as you rightly point out. 

As many questions remain on the use of somatostatin receptor PET/CT as an imaging modality to assess response to PRRT, we did not include it in our study design. Nevertheless it could be the object of future (pre-)clinical research.

Reviewer 2 Report

I appreciated the rigorous study protocol  investigating all the aspects of the combined administration of BSO and Lu-DOTATATE.

Results and discussion are exposed in a clear and rigorous manner.

It is no so frequent to read papers of this type.

Author Response

I would like to thank reviewer 2 for his/her comments on our research article.

Reviewer 3 Report

The manuscript described a combined therapeutic study design with GSH reducer and a PRRT agent Lu-177-DOTATE. Both in vitro and in vivo studies showed efficacy of this combination comparing with the either singular treatment or control. This combined strategy is novel and the results supported the authors' conclusion.

Author Response

I would like to thank reviewer 3 for his/her comments on our research article.